# Investigating the Functional Role of the Cysteine Residue in Dehydrin from the Arctic Mouse-Ear Chickweed *Cerastium arcticum*

**DOI:** 10.3390/molecules27092934

**Published:** 2022-05-04

**Authors:** Il-Sup Kim, Woong Choi, Ae Kyung Park, Hyun Kim, Jonghyeon Son, Jun Hyuck Lee, Seung Chul Shin, T. Doohun Kim, Han-Woo Kim

**Affiliations:** 1Advanced Bio-Resource R&D Center, Kyungpook National University, Daegu 41566, Korea; 92kis@hanmail.net; 2Research Unit of Cryogenic Novel Material, Korea Polar Research Institute, Incheon 21990, Korea; woong@kopri.re.kr (W.C.); insangkh@naver.com (H.K.); tuutoo92@dgmif.re.kr (J.S.); junhyucklee@kopri.re.kr (J.H.L.); ssc@kopri.re.kr (S.C.S.); 3Division of Bacterial Diseases, Center for Laboratory Control of Infectious Diseases, Korea Centers for Diseases Control and Prevention, Cheongju-si 28159, Korea; parkak1003@gmail.com; 4New Drug Development Center, Daegu-Gyeongbuk Medical Innovation Foundation, Daegu 41061, Korea; 5Department of Polar Sciences, University of Science and Technology, Incheon 21990, Korea; 6Department of Chemistry, Graduate School of General Studies, Sookmyung Women’s University, Seoul 04310, Korea; doohunkim@sookmyung.ac.kr

**Keywords:** dehydrin, Arctic mouse-ear chickweed, intermolecular disulfide bond, reactive oxygen species, dimerization, cysteine

## Abstract

The stress-responsive, SK_5_ subclass, dehydrin gene, *CaDHN*, has been identified from the Arctic mouse-ear chickweed *Cerastium arcticum*. CaDHN contains an unusual single cysteine residue (Cys143), which can form intermolecular disulfide bonds. Mutational analysis and a redox experiment confirmed that the dimerization of CaDHN was the result of an intermolecular disulfide bond between the cysteine residues. The biochemical and physiological functions of the mutant C143A were also investigated by in vitro and in vivo assays using yeast cells, where it enhanced the scavenging of reactive oxygen species (ROS) by neutralizing hydrogen peroxide. Our results show that the cysteine residue in CaDHN helps to enhance *C. arcticum* tolerance to abiotic stress by regulating the dimerization of the intrinsically disordered CaDHN protein, which acts as a defense mechanism against extreme polar environments.

## 1. Introduction

Arctic plants are exposed to various environmental stresses, such as freezing temperatures, drought and water deficiency, high ultraviolet radiation levels due to ozone depletion, nutrient depletion, salinity, and very short growing seasons [1,2].

Thus, to thrive at sub-zero temperatures, Arctic plants have developed diverse morphological and physiological adaptations for the different stages of growth development, such as long life cycles, extended primordial development of leaves and flowers, well-developed root systems, and efficient growth-development stages [3,4]. Dehydrins (DHNs) are a member of the late embryonic developmental abundance (LEA) protein family, which help plants during stress conditions, such as salinity, drought and low temperatures, which result in cell dehydration [5,6]. This protein was first identified in cottonseed at a later stage of embryonic development and was later found in other tissues and organs as well [7]. *LEA* genes have been significantly induced by abiotic stress and their overexpression in transgenic plants increased tolerance to a variety of abiotic stress sources [8,9]. A common feature of LEA proteins is their intrinsically disordered structure due to their amino acid composition, such as high glycine content and high hydrophilicity [10]. Based on their sequence similarity, LEA proteins are classified into three major groups (LEA I, LEA II, and LEA III) and each group has specific functions during dehydration [11]. The high accumulation of group II LEA proteins, including DHNs, is one of the prominent plant responses to cellular dehydration generated by abiotic stresses, such as cold, drought, and high salinity [12].

DHNs are a class of hydrophilic, thermostable cell rescue proteins with molecular masses ranging from 9 to 200 kDa [13,14] that are ubiquitous among various plants, algae, and cyanobacteria [12,15]. DHNs are very rich in glycine residues, whereas cysteine and tryptophan residues are lacking or under-represented [12]. A structural feature of DHNs is the existence of highly conserved sequence motifs called the K-, S-, and Y-segments [12]. The K-segment is a highly conserved lysine-rich motif (i.e., EKKGIMDKIKEKLPG) present in 1–11 copies near the C-terminal and it is the only conserved segment present in all DHNs [16,17]. Many DHNs also contain the serine-repeat S-segment formed by a stretch of 4–10 serine residues (LHRSGS4–10(E/D)3) and the tyrosine-rich Y-segment ((V/T)D(E/Q) YGNP) located near the N terminal [18,19]. In addition to the highly conserved domains, DHNs possess φ-segments defined as all the residues located between Y-, S-, and K-segments. The sequences and lengths of the φ-segments are not conserved compared with those in the other segments.

Arctic *Cerastium arcticum* Lange DHN (CaDHN) has been successfully expressed in yeast and *Escherichia coli* [20]. The overexpression of *CaDHN* in transgenic yeast confers tolerance against various forms of abiotic stress by improving redox homeostasis and enhancing fermentation capacity, especially at low temperatures (18 °C). Although most DHNs generally lack cysteine residue, CaDHN exhibits a single cysteine residue. In this study, we aimed to determine the role of the cysteine residue in CaDHN. For more detailed information, we performed a mutational analysis of CaDHN, which revealed its physicochemical properties via in vitro and in vivo assays. To the best of our knowledge, this is the first study that the unusual cysteine residue in CaDHN induces intermolecular disulfide bond-mediated dimerization of CaDHN, which plays an important role in the intrinsic and acquired tolerance of the transgenic yeast by modulating its function.

## 2. Results & Discussion

### 2.1. Disulfide Bond-Mediated Dimerization

CaDHN has five K-segment-like regions and one S-segment-like region [20]. Among the five K-segment-like regions, three were highly conserved relative to those in other DHNs. On the phylogenetic tree, CaDHN belonged to the DHN group close to LEA 2 (Appendix A). Interestingly, CaDHN has an unusual single cysteine residue (Cys143), which is very rare in other reported DHNs [21]. The cysteine residue is positioned at the φ-segment between the second and the third K-segments (Figure 1a). After the sodium dodecyl-sulfate polyacrylamide gel electrophoresis (SDS-PAGE) of the purified wild type (WT) protein, we observed two bands for the protein. The upper band was at approximately 80 kDa and disappeared after treatment with the reducing agent β-mercaptoethanol (β-ME). The lower band, at 40 kDa, did not migrate on the gel upon β-ME treatment (Figure 1b). To check whether the CaDHN formed an intermolecular disulfide bond, we constructed the mutant C143A, wherein an alanine residue was introduced instead of the cysteine. The mutant C143A protein exhibited only a single band in the non-reduced SDS-PAGE gel (Figure 1b) and not the upper band with high molecular weight as shown by the WT protein. On the native gel, both the WT and C143A proteins appeared as molecules of different sizes (Figure 2a). To confirm the existence of the intermolecular disulfide bond, we added different concentrations of dithiothreitol (DTT) and copper ions (as CuSO_4_) to both proteins and analyzed them on the native gel. DTT is a reducing agent that prevents disulfide bond formation between cysteine residues, and the oxidant copper ion is recognized as a transition metal ion that forces the formation of a disulfide bond between two cysteine residues by oxidation of their thiol groups [22,23]. As the DTT concentration increased, the upper (WT) band decreased in a dose-dependent manner and most of the protein migrated to the lower band at DTT concentrations above 8 mM; this position agrees with that of the C143A protein band on the SDS-PAGE gel. In contrast, increasing the copper concentration in the WT solution resulted in the migration of the small amount of the lower band migrate to the higher molecular weight upper band. This suggested that copper ions could trigger the formation of the disulfide bond in the WT protein, thereby moving to the upper band position due to the intermolecular dimerization of the protein. The C143A protein remained at the position of the lower band even in the presence of copper ions. These results suggested that CaDHN undergoes a molecular conformational change by disulfide bond-mediated dimerization, but not by metal-mediated dimerization between the protein molecules via amino acid residues, such as histidine [24,25,26]. The binding interaction between histidine residues and metal ions has been reported in citrus (*Citrus unshiu*) DHN [27]. The third non-conserved φ-segment motif containing the cysteine residue contains four histidine residues out of a total of 22 in the whole molecule. As shown in Figure 2a, in the C143A, there is no band shift after the addition of DTT and CuSO_4_. The upper band of the WT protein shifted completely to the lower band in the presence of 10 mM DTT. Thus, our results confirm that the WT protein could exist in the dimer form resulting from an intermolecular disulfide bond. Dimerization of DHN from plants has also been reported in the TsDHN-2 (Y2SK2 type) protein from *Thellungiella salsuginea* O.E. Schulz and the OpsDHN1 (SK3 type) from *Opuntia streptacantha* Lem. Interestingly, both DHNs dimerize differently in the structural interaction between their monomers. The TsDHN-2 protein has a potential dimerization state associated with a hydrophobic surface. However, in the OpsDHN1 protein, the region responsible for the dimer interaction is the histidine-rich motif that is located in the non-conserved segment between the first and the second K-segment [24,28].

### 2.2. Metal Binding Activity (MBA)

Some DHNs exhibit metal ion-binding properties, which have radical scavenging activity against cytotoxic hydroxyl radicals caused by metal-catalyzed reactions [27,29,30,31]. To investigate the metal-binding ability of the CaDHN, we tested their semi-quantitative MBA. Both the monomer and the dimer forms of CaDHN exhibited similar features regarding their MBA. As shown in Appendix A, both proteins were fully bound to the metal ions, Fe^3+^, Co^2+^, Ni^2+^, Cu^2+^, and Zn^2+^, but not to Mg^2+^, Ca^2+^, and Mn^2+^. These results were consistent with those obtained previously for citrus (*Citrus unshiu*) DHN [27]. The histidine residue in DHNs has been known to contribute to their MBA. CaDHN also has more abundant histidine residues than those of other proteins. To investigate DHN dimerization via other metal ions, we also performed native gel electrophoresis of both proteins in the presence of the same kind of metal ions as were used for testing the MBA. As a result, no size shift was observed with all seven tested metals, except for Cu^2+^ ions (Appendix A). Therefore, the in vitro assay showed that the CaDHN protein had MBA as mentioned above. Next, we tested in vivo *CaDHN*-mediated metal ion homeostasis using transgenic yeast. The metal-ion tolerance of *CaDHN*-expressing cells was improved compared with that of empty vector (EV) cells when yeast cells were challenged by stress induced by various metals including Co^2+^, Ni^2+^, Fe^2+^, and Zn^2+^ (Appendix A). On the other hand, the survival of *CaDHN*-expressing cells was similar to that of EV cells when treated with Mg^2+^, Ca^2+^, and Mn^2+^. These results for metal tolerance were almost consistent with the in vitro MBA of the protein (Figure 3). Interestingly, under copper stress, C143A *CaDHN* cells recovered more rapidly than WT *CaDHN* cells (Figure 3). According to a previous report, Ni^2+^, Cu^2+^, and Zn^2+^ bind to CuCOR15 protein, but Mg^2+^, Ca^2+^, and Mn^2+^ do not [27]. As seen in K*_n_*S-type DHNs, the SK_5_-type CaDHN binds to various metal ions [25]. Expression of *DHN2* and *DHN3* from *Brassica juncea* (L.) Czern. in transgenic tobacco enhanced its tolerance for heavy metals, such as Cd^2+^ and Zn^2+^ by attenuating lipid peroxidation and protecting cellular membranes after lower electrolyte leakage [32].

### 2.3. Yeast Stress Tolerance Assay

To investigate whether the dimerization of CaDHN protein affects its in vivo functional role, *CaDHN* (WT or C143A)-expressing transgenic yeast cells were developed. In transgenic yeast cells, the expression levels of both *CaDHNs* (WT and C143A) were identified by immunoblotting analysis, and the results were similar for both cells (Appendix A). Using the same transgenic yeasts, we examined whether *CaDHN* accumulation affects the acquired tolerance to various kinds of abiotic stress. Both transgenic cells were more tolerant than cells with EV alone when exposed to hydrogen peroxide (H_2_O_2_). Cells with C143A *CaDHN* were more tolerant to oxidative stress than those with WT *CaDHN* (Figure 4a). In general, although H_2_O_2_ induces the formation of disulfide bonds, which can directly oxidize cysteine residue, such oxidative cysteine modification triggers structural alterations in the target protein [33]. An irreversible change in DHN conformation resulted in changes in protein function [34,35]. In the presence of H_2_O_2_ during protein production, the CaDHN protein expressed in *E. coli* was found to be modified by hypo-oxidation [dioxidation (O2) and trioxidation (O3)] and carbamidomethylation (cam) (Appendix A). The modification of cysteine in the CaDHN monomeric state is irreversible. Our results imply that the increase in the amount of CaDHN monomers is not only induced by the modification of its cysteine residue through environmental influence or by exogenous stimuli, but also by a cellular redox protein. At high temperatures, the thermal tolerance of WT *CaDHN* cells was higher than that of C143A *CaDHN* cells, while the tolerance of WT *CaDHN* cells against cold stress was lower than that of C143A *CaDHN* cells (Figure 4b). The ability of C143A *CaDHN* cells to tolerate cold was better than that of WT *CaDHN* cells under various chilling conditions (Appendix A). A positive correlation has been found between the accumulation of DHN and the response of plants to abiotic stresses. For example, overexpression of the *Arabidopsis thaliana* DHN genes (LTI29 and LTI30) was associated with increased tolerance to stress from freezing, dehydration, and chilling [36]. In addition, overexpression of the *C. unshiu* COR19 gene in transgenic tobacco reduces ion leakage and lipid oxidation under cold and freezing conditions, thereby increasing cold stress tolerance [37,38]. Therefore, these results indicate that the monomer form of the CaDHN protein leads to enhanced physiological tolerance of the cell for low temperatures and H_2_O_2_, but not for high temperatures.

In summary, CaDHN has a unique single cysteine residue, which plays a role in its dimerization via an intermolecular disulfide bond. CaDHN exhibits biological functions including redox and metal ion homeostasis. Our results suggest that regulation of the cysteine-mediated disulfide bond of CaDHN following the cellular redox state could improve the tolerance of polar plants to abiotic stress through conformational changes.

## 3. Materials and Methods

### 3.1. Cloning, Expression and Purification

Construction of recombinant *CaDHN* in *E. coli* was performed as reported previously [20]. In brief, total RNA from *C. arcticum* L. (*Caryophyllaceae*) leaves collected from Brogger-Halvoya, Svalbard was isolated using the RNeasy Plant Mini kit (Qiagen, North Rhine-Westphalia, Germany) according to the manufacturer’s instructions and cDNA was synthesized using reverse transcriptase. The *CaDHN* gene was amplified by polymerase chain reaction with a specific primer set harboring restriction enzyme sites represented in Appendix A. *CaDHN* gene was introduced between the *Eco*R*I* and *Xho*I sites within the pET30a vector (Novagen, Reno, NV, USA) (Appendix A) using the sequence- and ligation-independent cloning method. A tobacco etch virus (TEV) protease recognition sequence was inserted in front of the *CaDHN* gene. During the construction of recombinant CaDHN protein, we recognized that the N-terminal long tag contained within the pET30a vector was able to enhance the CaDHN overexpression. The N-terminal tag is composed of 52 amino acids corresponding to about 6 kDa. In detail, the sequence of the N-terminal long tag is MHHHHHHSSGLVPRGSGMKETAAAKFERQHMDSPDLGTDDDDKAMADIGSEF. The recombinant plasmid was transformed into *E. coli* BL21 (DE3) strain. For purification, the pelleted cells were suspended in the lysis buffer containing 20 mM Tris-HCl at pH 8.0, 200 mM NaCl and 10 mM imidazole, and lysed by sonication. After centrifugation, the supernatant was loaded onto nickel-charged affinity resin and the target protein was washed and eluted with the wash buffer containing 20 mM Tris-HCl at pH 8.0, 200 mM NaCl, 30 mM imidazole and the elution buffer containing 20 mM Tris-HCl at pH 8.0, 200 mM NaCl, and 300 mM imidazole, respectively. Next, TEV protease (New England Biolabs, Ipswich, MA, USA) was added to the solution and dialyzed in dialysis buffer (20 mM Tris-HCl at pH 8.0, 200 mM NaCl and 10 mM imidazole) at 4 °C overnight. A second nickel-charged affinity chromatography step was performed to remove uncleaved CaDHN and the N-terminal tag peptide. The mutant gene of C143A was prepared by Site-Directed Mutagenesis using PCR (Appendix A). The predicted molecular weight of the CaDHN protein was calculated to be 29,509 Da.

### 3.2. Native Gel Electrophoresis

Native gel electrophoresis was run on 10% polyacrylamide gels. The electrode buffer contained 25 mM Tris base and 19 mM glycine. All native gels were run in an ice bath to dissipate heat generated during the runs. The protein bands were visualized using Coomassie Brilliant Blue R-250.

### 3.3. Development of Yeast Expression Vector and Transgenic Yeast

For the development of the yeast expression vector bearing the *CaDHN* gene, the target gene was subcloned into the digested p426GPD vector using the glyceraldehyde-3-phosphate dehydrogenase (*GPD1*) promoter (Euroscarf, Oberursel, Germany) (Appendix A). The resulting plasmid was named p426GPD:CaDHN_WT. The mutant gene of *CaDHN*_C143A was prepared by the PCR-dependent Site-Directed Mutagenesis kit (New England Biolabs) using the p426GPD::CaDHN_WT backbone, according to the manufacturer’s protocols (Appendix A). The resulting plasmids (p426GPD::CaDHN_WT and p426GPD::CaDHN_C143A) were used to transform a conventional *Saccharomyces cerevisiae*. Haploid *S. cerevisiae* BY4741 (Euroscarf) cells grown at 28 °C overnight in a nutrient-rich Yeast Extract-Peptone-Dextrose (YPD) broth medium containing 1% yeast extract, 2% peptone, and 2% dextrose were reinoculated into a fresh YPD broth medium, and then incubated for 4 h at 28 °C with shaking (180 rpm) until the optical density of the culture solution at 600 nm (A_600_) reached approximately 1.0 (1 × 10^7^ cells/mL) to produce early exponential cells. The resulting yeast cells were then transformed with the two target plasmids (p426GPD::CaDHN_WT, p426GPD::CaDHN_C143A) by using the PEG/LiCl method reported previously [39]. The transformed cells were plated on a minimal nutrient agar medium (0.67% yeast nitrogen base without amino acids and with ammonium sulfate, 0.192 % yeast synthetic drop-out medium supplement without uracil (Sigma-Aldrich, Burlington, MA, USA), 2% glucose, and 1.5% agar). The plates were incubated at 28 °C for 3–4 days. Positive colonies grown in a medium lacking uracil were selected and used for subsequent experiments.

### 3.4. Stress Response Assay in Yeast Cells

Yeast cells grown in the YPD broth medium until the mid-log phase (A_600_ = 2.5; 1 × 10^9^ cells/mL) were exposed to different concentrations of H_2_O_2_ (2 mM and 3 mM), zinc chloride (100 mM ZnCl_2_), copper sulfate (10 mM and 20 mM CuSO_4_), cobalt chloride (80 mM CoCl_2_), nickel sulfate (80 mM NiSO_4_), ferric chloride (80 mM FeCl_2_), manganese chloride (80 mM MnCl_2_), calcium chloride (250 mM CaCl_2_), and magnesium chloride (250 mM MgCl_2_) for 1 h with shaking (180 rpm). They were then serially diluted to 10^−4^ with fresh YPD broth medium. Five microliters of this diluted solution were then loaded onto YPD agar (YPD plus 1.5% agar) plates, which were incubated for 3 days at 28 °C and then photographed. On the other hand, for extreme temperatures, yeast cells of the same phase were serially diluted to 10^−9^ with fresh YPD broth, loaded onto YPD agar plates and incubated for 6 days at 40 °C, and for 11 days at 15 °C, respectively.

## Figures and Tables

**Figure 1 molecules-27-02934-f001:**
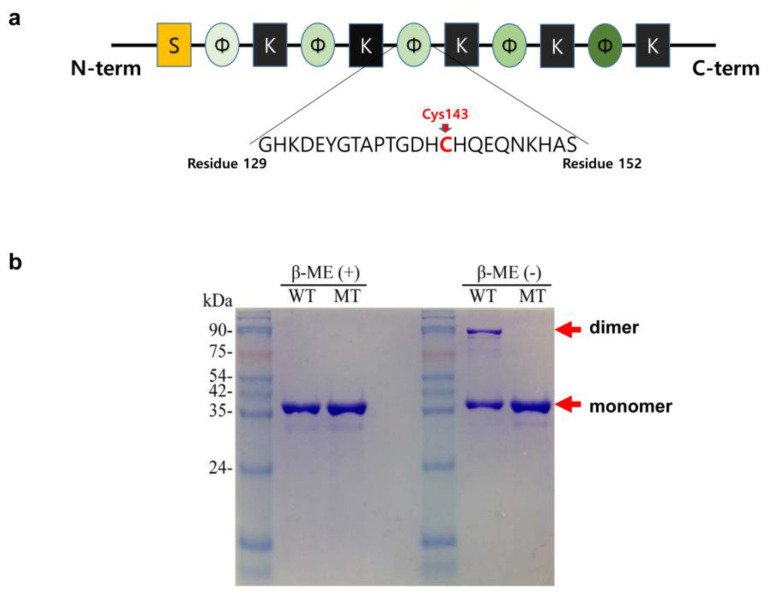
Dimerization of *Cerastium arcticum* Lange dehydrin (CaDHN) by the unique cysteine. (**a**). Scheme of the cysteine position in the CaDHN. The CaDHN protein containing 270 amino acids was composed of five K segments, a conserved S segment and five non-conserved φ-segments. The unique cysteine within the φ-segment is highlighted in red. (**b**). Purified CaDHNs analyzed on Coomassie blue were stained on an SDS-PAGE gel. WT, the wild type protein; MT, the mutant C143A protein; β-ME(+), protein treated with 10 mM of β-mercaptoethanol. Each gel lane was loaded with 1 μg of the protein.

**Figure 2 molecules-27-02934-f002:**
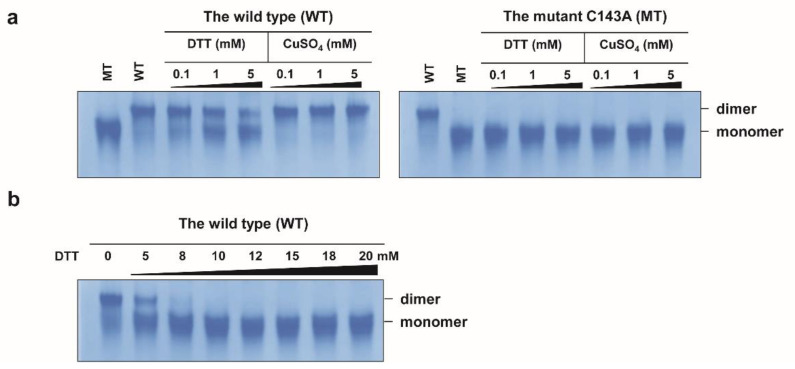
Native gel analysis for dimerization via disulfide bond of the purified *Cerastium arcticum* Lange dehydrins (CaDHNs). (**a**) Both wild type (WT) and mutant C143A (MT) proteins treated with different concentrations of dithiothreitol (DTT) and copper ions (CuSO_4_). (**b**) DTT-treated WT protein (0–20 mM). Each gel lane was loaded with 1.5 μg of the protein.

**Figure 3 molecules-27-02934-f003:**
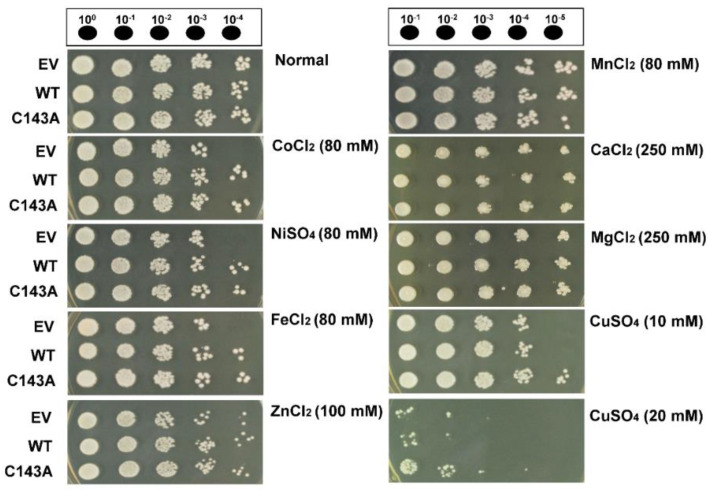
A metal-mediated stress response assay of *CaDHN* (WT and C143A)-expressing transgenic yeast on plates. Cells transformed with an empty vector (EV; p426GPD)) were used as control. Mid-log yeast cells were exposed to stressors for 1 h with shaking (180 rpm), serially diluted with a fresh YPD broth medium and spotted onto YPD agar plates. Five microliters were used for a spotting assay. The plates were incubated for 3–4 days at 28 °C and photographed. The results were representative of at least three independent experiments conducted under identical conditions.

**Figure 4 molecules-27-02934-f004:**
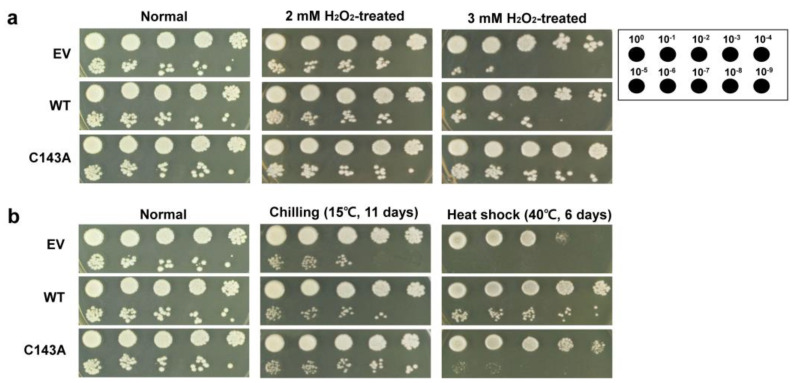
Cell viability of transgenic yeast against oxidative and thermal stress. An oxidative stress tolerance assay (**a**); thermal stress tolerance assay (**b**). Stress responses of *Saccharomyces cerevisiae* BY4741 cells expressing the *CaDHN* gene were determined by the spot assay. The cells transformed with an empty vector (EV) were used as a control. Mid-log yeast cells were serially diluted to 10^–9^ with YPD broth medium and 5 μL of the diluted solution were spotted onto YPD agar medium in the absence and presence of hydrogen peroxide (H_2_O_2_). The plates were incubated at 28 °C during the time indicated and then photographed. For H_2_O_2_-mediated spotting assay, the plates were incubated for 3–4 days at 28 °C. The results were representative of at least three independent experiments conducted under identical conditions.

## Data Availability

Data is contained within the article and Supplementary Material.

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
