# Peer review of "Investigating the Functional Role of the Cysteine Residue in Dehydrin from the Arctic Mouse-Ear Chickweed Cerastium arcticum"

_molecules, 2022, doi:10.3390/molecules27092934_

Round 1

Reviewer 1 Report

The authors describe an intermolecular disulfide bond between two dehydrin molecules that might be involved in stress response. Mutation of a cysteine to an alanine prevented disulfide bond formation  in the presence of DTT and also Cu2+-concentrations that otherwise seem to promote protein dimerization. Overall the report contains interesting data, but is rather sloppily and confusingly written despite so many authors. Several connections are difficult to find and some explanations are not satisfying. The sequence of individual chapter is also not logical. 2.3 should follow 2.1 and this all should be reorganized and structured in a superior way. In addition, several general stress effects are just discussed on the surface and poorly correlated to the observations by the authors and how these experiments might be continued in the future. A true outlook is completely missing, so what to do next? Did all authors read this and were happy with the presentation of these data? Quite frankly, I am not.

In addition, many minor and major language errors, even the keywords contain mistakes, arctic not: Arc tic. Please, have a native speaker or editor correct the language! In a few cases, this is awkward.

Specific critical comments:

Question: How exactly does Cu2+ promote DHN dimerization, this is not explained in detail. Please comment.

Line 40: Dehydrins (DHN)… the abbreviation should at least be explained once.

Line 110/142: Citrus spec. not citrus… is this Citrus unshiu?? Please be consistent!

Line 135: There is a sudden break in the logic: Chapters 2.2. and 2.3 need to be exchanged. The authors move from the C143A observation to binding affinity that is based on histidine residues, why this? The reader expects now a deeper investigation of the consequences of the C143A mutationàChapter 2.3. Lines 135-147 should be placed immediately before lines 186-199 and this should be one chapter, in this case chapter 2.3. So, line 186 can actually be deleted and line 147 continued with: Hence… Otherwise this is confusing and redundant.

Line 136… resulting in radical scavenging activity…

The response to the different metal ions is quite different, and divided into two groups protecting or not protecting yeast cells at low density, differences in the case of Co, Ni, Fe, no difference in the case of Zn, Mn, Ca, and Mg.  The only effect of the mutation is in the case of Cu ions. To my opinion, in most cases the concentration of the metals is extremely high, more than 10 mM is quite arbitrary. Please comment! Also, please correlate the response to CuCOR15 with your data. What exactly is the connection?

Line 209/210: Please rephrase first part (be specific) and delete the second part of the sentence, unless there is any clear statement how this speculation can be practically achieved.

Figure 1: A. Scheme of the cysteine position in the CaDHN protein.

Figure 2: Of course there are some effect in the mutant C143A but: again, the legend should explain the Figure. While the mutant protein does not show dimerization Increasing concentrations of DTT up to 5 mM indicate the dimeric state of the protein, whereas higher DTT concentrations result in a monomeric protein. But how much protein is loaded onto each lane? What is the ration of protein versus DTT? This is relevant!

Figures 3/4: how often were the assays performed?  If n=1, repeat at least 2 x so n=3. Independent replicates! What does 10o to 10-9 mean, is this cell dilutions? The legend should explain the data shown. lines 184/202: the English language requires editing. Better: delete this sentence and rewrite the legends in both cases i.e. point out the most relevant data and results of the individual experiment!

Figure S2: I do not see the difference in the case of Cu/shift. Please mark clearly what you mean. Please clearly mark the difference between mutant C143A and WT, if there is any.

Figure S9: Why chilling 10 days and 11 days at 15oC, why not a standard time frame from 5 to 7 to 9 to 11 days. How often was the experiment performed n= 1? This is insufficient please repeat. Details should be explained in the legend.

Line S183: was cut (!) and free thiols were blocked…

Several minor syntax errors throughout the manuscript! Carefully check and edit.

Reviewer 2 Report

The following manuscript entitled “Investigating the functional role of the cysteine residue in dehydrin from the Arctic mouse-ear chickweed Cerastium arcticum by Kim et al. explains the role of cysteine residue. Overall the manuscript is written very well. However, some major issues need to be addressed before taking any decision.

Similarity Index

Plagiarism in the current manuscript is 26% which is way too much than the permissible limit.

Materials and Methods

Section 3.1. This section is very confusing please rewrite this section clearly so that one should understand it easily.

“In brief, C. arcticum L. (Caryophyllaceae) was collected from Brogger-Halvoya, Svalbard. For what reason? please mention!

 Section 3.4. “Yeast cells grown were exposed to each concentration….” Please mention the concentrations and their selection criteria.

Results and Discussion

Section 2.1. “Treatment with the reducing agent β-mercaptoethanol, and other treatments such as DTT and CuSO4.” It would be better if the author should also mention these in the Materials and Methods section and the concentrations of each treatment.

General comments:

  • There are so many problems with English grammar and sentence buildup.
  • Avoid formatting mistakes.

Round 2

Reviewer 1 Report

The authors have addressed all critical comments. The manuscript is in a much better shape now. I is clear and straight forward. Besides the improved handling of some Figures specifically the minor syntax errors have been removed. Fine with me now.

Two minor suggestions in the case of Supplemental Files:

  1. S1 Please add some more information to the legend, remember this should be self-explanatory and the description is simply not detailed enough. What is marked in Red? How can individual colors be described, differentiated? What about the sequences that are not part of colored cluster. Explain in more detail!
  2. S2: my suggestion is instead of a and b, divide this Figure in a,b,c,d and again explain in more details what is shown. Supplementary should have the same quality and information as a Figure in the manuscript.

Reviewer 2 Report

The authors have improved their article substantially.
